# Biomolecules Related to Rotator Cuff Pain: A Scoping Review

**DOI:** 10.3390/biom12081016

**Published:** 2022-07-22

**Authors:** Nikolaos Platon Sachinis, Christos K. Yiannakopoulos, Byron Chalidis, Dimitrios Kitridis, Panagiotis Givissis

**Affiliations:** 1“Georgios Papanikolaou” Hospital, 57010 Thessaloniki, Greece; byronchalidis@gmail.com (B.C.); dkitridis@gmail.com (D.K.); pgivissis@gmail.com (P.G.); 2School of Physical Education & Sports Science, National & Kapodistrian University of Athens, 15772 Athens, Greece; ckyortho@gmail.com

**Keywords:** biomolecules, shoulder, rotator cuff, pain, IL-1b, VEGF, MMP, TNF-a, IL-8, IL-10

## Abstract

The pathophysiology of pain in patients suffering from rotator cuff (RC) tendinopathy or tears has been examined in various ways. Several molecules from tissue samples taken from the subacromial bursa, supraspinatus tendon, glenohumeral joint fluid, and synovium as well as from peripheral blood have been investigated. This article explores these studies, the assessed biomarkers, and groups their results according to the status of tendon integrity (tendinopathy or tear). Through a structured PubMed database search, 9 out of 658 articles were reviewed. Interleukins, mostly IL-1b and its antagonist, IL-1ra, matrix Metalloproteinases (MMPs), the vascular endothelial growth factor (VEGF) and TNF-a are biomarkers directly searched for correlation to pain level. Most studies agree that IL-1b is directly positively correlated to the degree of pain in patients with RC tendinopathy, especially when the examined sample is taken from the subacromial bursa. VEGF, and TNF-a have been related to shoulder pain preoperatively and TNF-a has also been linked with sleep disturbance. Further studies pointing to more biomarkers taken from the subacromial bursa or tendon directly relating to pain degree are warranted.

## 1. Introduction

The rotator cuff (RC) of the shoulder consists of the subscapularis tendon, the supraspinatus, the infraspinatus, and the teres minor muscles. Rotator cuff pathology affects a large percentage of the population and in particular the elderly [1]. Patients may experience debilitating pain and reduced function. The natural history and molecular pathology of cuff illness are still not fully understood. Historically, the concept of intrinsic and extrinsic tendon mechanics has been explored and debated [2]. Failure of cuff homeostasis in degenerative disease and pain is a process that involves a plethora of biomolecules.

Although the nociception histological and molecular pathway has been described [3], it is not clear why patients suffering from the same macroscopic tendon pathology experience pain in such varying degrees. Inflammatory and angiogenic cytokines have been linked to RC tendinopathy and tear as interleukins upstream regulate the inflammatory pathway in these tissues [4,5]. Matrix metalloproteinases (MMPs) may be also linked to inflammation and tendon wear as they can degrade proteins of the extracellular matrix and also upregulate the expression of cytokines and chemokines [6].

Many other biomarkers have been also detected in bursa and tendon tissue of patients and cadavers with degenerative or torn rotator cuff and have been reviewed [2,7]. Matrix components, growth factors, enzymes, and cytokines play a variable role in the pathogenesis of rotator cuff disease, from tendinopathy to tear.

Due to the plethora of biomolecules and the heterogeneity of investigations performed so far, we conducted a scoping review to search human studies that examined biomolecules found in synovial fluid, bursal, or tendon tissues, in patients suffering from rotator cuff-related pain.

## 2. Methods

### 2.1. Review Type

We decided to conduct a scoping review versus a systematic review, following Munn et al.’s recommendations [8]. So far, no consensus exists on which biomolecules should be investigated and which tissue should be harvested, in relation to rotator cuff-related pain. Regarding the condition of the rotator cuff (tendinopathy or full tear), it is unknown how each molecule concentration correlates to nociception. Our research question was to identify molecules found in shoulder tissue responsible for cuff-related pain. For non-traumatic cases, tendon degeneration is a continuous progressive disease. However, for review purposes, we categorized this degenerative process as cuff tendinopathy (non-perforating) or cuff tear.

### 2.2. Studies Retrieval

All studies presented were found through a systematic approach via the PubMed website. Only human studies written in English, of any year, were searched. Firstly, we searched for previous relative studies and reviews and identified a list of biomolecules that could be related to our research question. Search in title and abstract began with the keywords “shoulder” AND “biomolecules” retrieving 12 results (1 review) and “shoulder AND “biomarkers” retrieving 557 results (73 reviews). A list of abbreviations of the biomolecules is presented in Table 1.

Then, after finding the related molecules, we conducted a final search in March 2022 by using keywords in the following way: ((Shoulder[Title/Abstract]) AND (pain[Title/Abstract])) AND ((IL[Title/Abstract]) OR (CD45[Title/Abstract]) OR (CD206[Title/Abstract]) OR (KA1[Title/Abstract]) OR (mGluR2[Title/Abstract]) OR (MMP[Title/Abstract]) OR (NMDAR1[Title/Abstract]) OR (PGP9.5[Title/Abstract]) OR (TNF[Title/Abstract]) OR (VEGF[Title/Abstract])). We retrieved 89 results from the last search having a total of 658 results. After removing duplicates and reviews, we screened 550 studies. We excluded 526 studies for not being related to our review or not performed in humans, and assessed 24 studies. Studies were thoroughly analyzed and checked in the discussion and reference section.

We finally excluded 15 studies for not having a specific biomolecule directly associated with cuff-related pain, finally leaving 9 studies for analysis in our review. The Preferred Reporting Items for Systematic reviews and Meta-Analyses (PRISMA) flow diagram is shown in Figure 1 [9]. A summary of the studies investigated and their key findings are found in Table 2. The PRISMA extension for Scoping reviews checklist can be found in Appendix A.

## 3. Results

### 3.1. Pain and Rotator Cuff Tendinopathy

Few studies directly investigated the relationship of biomolecules with pain in patients suffering from RC tendinopathy. The scientific team of Gotoh, Hamada, Yanagisawa, and their colleagues from 1998 to 2002 produced considerable work regarding the search for biomolecules that attributed to the feeling of pain in that subset of patients [10,11,12,13]. Biomolecules related to angiogenesis and inflammation were specifically studied by this team.

Angiogenesis is an important feature in inflammatory diseases. In shoulder tendinopathy, the inflammatory response is mediated by several biomarkers. The vascular endothelial cell growth factor (VEGF) has a powerful angiogenic and mitogenic ability that targets specifically the vascular endothelial cells [19]. Yanagisawa et al. studied 50 patients with rotator cuff disease, including 12 patients with subacromial bursitis, 12 patients with a partial-thickness tear, and 26 patients with full-thickness cuff tear (not specifying the tear size) [11]. Motion pain, nocturnal pain and spontaneous pain were recorded via a questionnaire that used a visual analogue scale (VAS). The authors collected synovial specimens from the subacromial bursa during surgery. There were 41/50 patients that complained of shoulder pain and almost all (39/41) showed increased mRNA VEGF expression. Only one out of nine patients without pain showed elevated VEGF levels. Therefore, a statistically significant positive correlation between pain and VEGF was demonstrated.

Gotoh et al. in 2001, took synovial tissue specimens from 39 patients undergoing shoulder surgery due to subacromial bursitis (10 patients), partial-thickness RC tear (9 patients) and full thickness RC tear (20 patients) [12]. Pain was evaluated with VAS, before the operation. The mRNA levels of IL-1b and two forms of IL-1ra were examined by reverse transcriptase- polymerase chain reaction (RT-PCR). The authors also detected the cells producing these molecules by immunohistochemistry and in situ RT-PCR. However, the authors did not subgroup their results according to the type of cuff tendinopathy/tear. Their results showed that in the bursa samples, the icIL-1ra mRNA in synovial lining cells was up-regulated, compared to the sIL-1ra in sublining cells, in line with the overexpression of IL-1b mRNA that was made by both cell types. IL-1b and its antagonists, sIL-1ra and icIL-1ra, were positively significantly correlated to the degree of pain.

### 3.2. Pain and Rotator Cuff Tear

An RC tear has been described as perforating when the full thickness of the tendon has been torn and the subacromial space directly communicates with the glenohumeral joint space [10]. Some studies have therefore merged patients with partial thickness tear and patients with subacromial bursitis in one group and separated patients having full thickness tear in another group (perforating tear group) [10,13].

Substance P is a molecule involved in the transmission of pain and its release from small sensory neurons may mediate neurogenic inflammation [20]. Gotoh et al. in 1998 investigated the subacromial bursa tissue regarding the correlation of substance P with shoulder pain in patients grouped according to the type of tendon degeneration (non-perforating vs. perforating tear). Immunochemistry tests revealed that substance P levels were greater in the non-perforating group, whose patients also reported more pain. The authors of this study advocated that these differences in substance P may be attributed to the greater amount of bursa found in the non-perforating group. It was hypothesized that continuous impingement and subsequent full thickness tear may progressively wear out and denervate the bursa [10].

Several IL biomolecules and mostly IL-1b have been studied in relation to RC pain. Gotoh et al. [12] who had demonstrated a positive correlation between IL-1b and its antagonists in the subacromial bursa of patients with RC pain, also investigated the same biomarkers in the synovial tissue of the glenohumeral joint. Specimens were taken from the rotator interval area of 35 patients undergoing bursectomy for perforating and non-perforating tears. Interestingly they found higher levels of IL-1b and IL-1ra in the glenohumeral synovium of patients with perforating tears but their values were inversely correlated with pain. The authors summarized that synovitis of the glenohumeral joint did not produce pain in rotator cuff disease.

IL-1b has been also investigated in other clinical studies. Siu et al., in 2013 took subacromial fluid samples from 68 patients with RC tear who were grouped according to diabetic status. Fluid was tested for IL-1b levels with an enzyme-linked immunosorbent assay (ELISA). It was found that IL-1b levels were positively correlated with the degree of pain. Furthermore, diabetic patients had significantly higher IL-1b levels and associated pain when compared to the non-diabetic group.

Along with IL-1b, other interleukins have also been studied. Okamura et al. utilized ELISA to test for IL-1b, IL-6, and IL-8 in the glenohumeral joint fluid of 38 patients undergoing RC surgery. Cuff tears were sub-grouped according to the Cofield classification [21]. Pain was measured by VAS during rest, at motion, and at night-time. The RC tear size was inversely correlated with VAS at rest. Pain at rest was also strongly correlated with IL-8. A positive trend was shown for IL-1b (r = 0.298, *p* = 0.069), but was not significant.

Shih et al. studied the glenohumeral joint fluid of 42 patients with a chronic RC tear, defining chronic as a tear that was at least 3 months old [17]. Rotator cuff severity was noted as partial- thickness tear, non-massive full thickness tear and massive tear. ELISA testing detected that IL-1b levels were positively correlated with VAS and other functional scores [17].

Sleep disturbance related to pain in RC tear cases has also been examined in relation to IL-type biomolecules. Cho et al. made a 3-arm trial of 63 patients, including 21 patients with RC tear and sleep disturbance (SD group), 21 patients with RC tear but with normal sleep (NS group), and 21 asymptomatic patients (control group) [18]. Peripheral blood samples were taken from patients and studied via ELISA kits for IL-1a, IL-1b, IL-2, IL-6, IL-8, IL-10, and TNFa. The authors found that IL-8 and IL-10 were significantly higher in the SD group when compared to the control group, but not when compared to the NS group.

The above study also found that TNF-a levels were significantly higher in the SD group when compared to both other groups [18] and therefore an association between TNF-a and pain-related sleep disturbance in patients with RC tear was demonstrated. Another study identified that cytokines and TNF-a were increased along with melatonin levels [22]. Although it is not been directly studied so far, a relation between the level of pain during night sleep and melatonin levels could be hypothesized in patients with RC tear,

The levels of TNF-a along with other biomarkers in association to RC-related pain has also been studied by Dean et al. [15]. Supraspinatus tendon biopsies were taken from 18 patients having RC tendinopathy (excluding full thickness RC tears). Subjects were grouped according to pain level and operative status; one group consisted of nine patients who had persistent pain and were ready to undergo surgery and the other group included nine who had complete pain relief after a shoulder operation. Histology, immunochemical, and ex vivo analysis of tenocyte explants from a separate cohort was performed to detect the gene expression of inflammatory and glutaminergic markers. The study demonstrated increased levels of metabotropic glutamate receptor 2 (mGluR2), kainate receptor 1 (KA1), protein gene product 9.5 (PGP9.5), CD206 (macrophage marker), and CD45 (pan-leucocyte marker) in patients with persistent pain. Moreover, the expression of mGluR2, N-methyl-D-aspartate receptor (NMDAR1), KA1, CD45, CD206, and tumour necrosis factor alpha (TNF-α) genes was significantly increased in disease-derived cells vs. control cells [15].

Although the role of MMPs, which are a family of 24 zinc-dependent endopeptidases, in tissue remodeling after injury has been documented, their overactivity may cause progressive wear of the tendon matrix [23]. Shih et al. in 2017, apart from interleukins also examined MMP-1 and MMP-13 in their cohort of 42 patients with chronic RC tear. MMPs were not found to be directly correlated with pain. However, their levels were significantly higher in massive full-thickness tears. This was in accordance with other studies relating to tendon wear and MMPs expression. Castagna et al. examined MMP 1, 2, and 3 and found their levels altered not only at the edge of supraspinatus tears but also in regions far from the torn tendon [24].

## 4. Discussion

This study has identified several biomolecules that are associated with rotator cuff-related pain. Substance P, IL-1a, IL1b, IL-2, IL-6, IL-8, IL-10, TNF-a, and VEGF are involved in the pathophysiologic mechanism of subacromial inflammation. Human studies have narrowed analysis mostly on cytokines IL1-b and TNF-a, which have been positively correlated with nociception [12,13,16,17,18]. Sleep disturbance related to pain seems to be also related to these cytokines. Expectedly, the presence of inflammation induces pain. VEGF, being responsible for angiogenesis is key to bursa inflammation [11]. It has been found to be positively correlated to nociception and is, therefore, another molecule-candidate for further research.

No tissue so far has been recommended as the ideal candidate for retrieving these molecules from humans. The subacromial bursa is a tissue that has a direct topological relation to the rotator cuff and undergoes inflammatory changes in several stages of tendinopathy. Literature has supported the presence of pain-related molecules in that tissue [10,11,12]. Bursa may also be more easily harvested than the suprascapular tendon, with the caveat of being non-existent in some rotator cuff tear cases [25]. Synovial fluid may contain a sufficient amount of cytokines for analysis but does not have direct contact with the subacromial space, apart from full thickness tear cases; we are also unaware of any studies tracing VEGF in this fluid. Therefore, we encourage harvesting bursa samples for future studies and analysis of more molecules.

This scoping review has several limitations. We have examined only human studies and have not taken into account animal-model studies. However, our goal was to narrow down the biomolecules found in humans and target which tissues would provide these molecules consistently, in order to encourage further human studies and future systematic reviews, focusing on specific biotargets. Also, we only searched one database and we may have missed human studies not published in the PubMed database. We do not know if the included studies are suitably statistically powered for their investigations, as none of them had an a-priori power analysis performed.

## 5. Conclusions

The subacromial bursa has been demonstrated to produce two major biomarkers of RC-related pain, IL-1b, and VEGF. Biopsies of not fully torn supraspinatus tendons support the increased expression of inflammatory and angiogenic molecules from that tissue.

The published results of synovial fluid analyses are mixed, especially when the fluid directly communicates with the subacromial space. Future studies should focus on identifying more biomarkers directly related to pain, taken from the subacromial bursa. Since TNF-a has been shown to be linked with sleep disturbance, its bursa levels and relation to pain should be further clarified. Melatonin levels may be also examined in relation to pain levels during sleep in patients suffering from RC tendinopathy or tear. Antagonists of these biomarkers may be valuable in reducing the pain in this shoulder patient population.

## Figures and Tables

**Figure 1 biomolecules-12-01016-f001:**
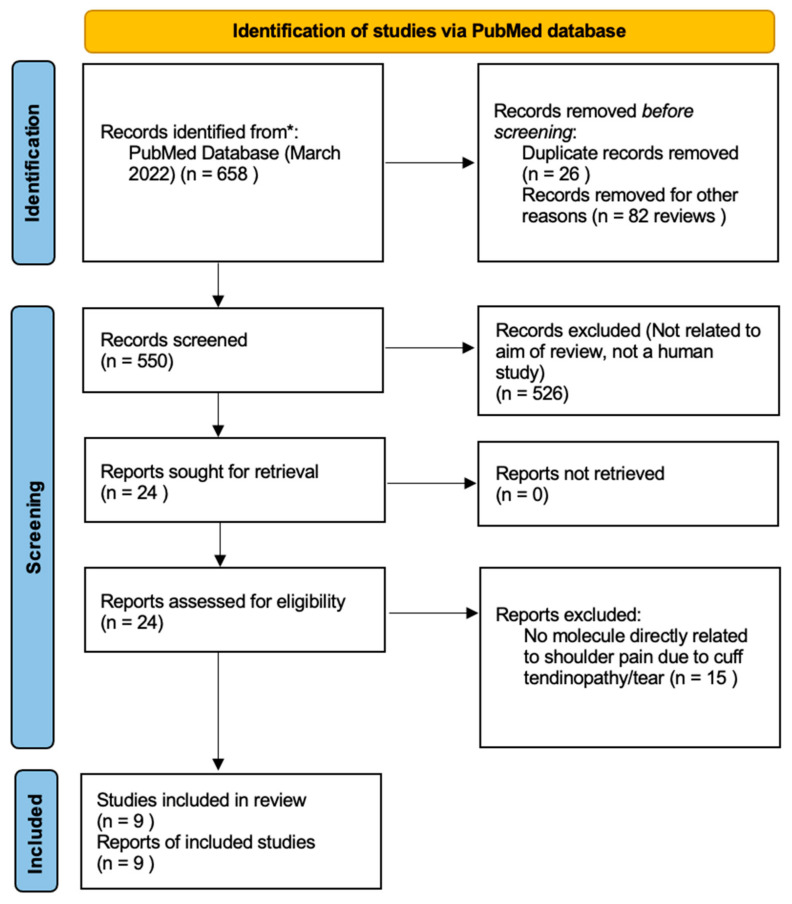
PRISMA flow diagram of this scoping review. * Two stage search was conducted in order to identify specific biomolecules related to rotator cuff related pain.

**Table 1 biomolecules-12-01016-t001:** List of abbreviations regarding biomolecule types.

Biomolecule Type	Abbreviation
Protein tyrosine phosphatase, receptor type, C	CD45
Mannose receptor C type 1	CD206
Interleukin	IL
Interleukin 1 antagonist receptor	IL-1ra
kainate receptor 1	KA1
metabotropic glutamate receptor 2	mGluR2
Matrix metalloproteinases	MMP
N-methyl-D-aspartate receptor	NMDAR1
protein gene product 9.5	PGP9.5
Tumor Necrosis Factor	TNF
Vascular endothelial growth factor	VEGF

**Table 2 biomolecules-12-01016-t002:** Human studies directly investigating biomolecules in patients suffering from pain due to rotator cuff tendinopathy/tear.

Study	Sample Size	Tissue Examined	Biomarker Measured	Key Findings
M. Gotoh et al. 1998 [10]	38 patients (19 non-perforating group; 18 perforating group) vs. 9 cadaver specimens	Subacromial bursa tissue	Substance P	Pain greater in the non-perforating group Substance P levels greater in the non-perforating group
K.Yanagisawa et al 2000 [11]	50 patients (12 with subacromial bursitis; 12 with partial-thickness tear; 26 with full-thickness cuff tear)	Subacromial bursa	VEGF	Positive relation between VEGF presence and pain 10 VEGF (−) 40 VEGF (+)
M. Gotoh et al. 2001 [12]	39 patients,10 with subacromial bursitis; 9 with partial rotator cuff tear; 20 with full tear)	Subacromial bursa	Inflammatory cytokines: mRNA levels of IL-1b, IL-1ra	mRNA expression levels of the cytokines were significantly positively correlated with the degree of pain
M. Gotoh et al. 2002 [13]	35 patients (16 with non-perforating tears; 19 with perforating tears)	Synovial tissue of glenohumeral joint	mRNA levels of IL-1b, IL-1ra	Cytokine-mRNAs greater in perforating tears than in non-perforating tears Cytokine-mRNAs inversely correlated with the degree of pain Cytokine-mRNAs in the synovium of the glenohumeral joint contribute less to the shoulder pain in rotator cuff diseases
K. Siu et al. 2013 [14]	68 patients with rotator cuff tear (23 diabetic and 45 nondiabetic patients)	Subacromial synovial fluid	IL-1b	Diabetic patients increased IL-1b IL-1b levels correlated with VAS score
B. Dean et al. 2015 [15]	18 patients with supraspinatus tendinopathy, excluding full thickness tears (9 patients with persistent pain before surgery; 9 operated and with no pain at follow-up)	Supraspinatus tendon	mGluR2; KA1; PGP9.5; CD206 (macrophage marker); CD45 (pan-leucocyte marker); mGluR2; NMDAR1; TNF-α); VEGF	Increased levels of metabotropic glutamate receptor 2 (mGluR2), kainate receptor 1 (KA1), protein gene product 9.5 (PGP9.5), CD206 (macrophage marker) and CD45 (pan-leucocyte marker) in persistent pain group Increases in the expression of mGluR2, N-methyl-D-aspartate receptor (NMDAR1), KA1, CD45, CD206, and tumor necrosis factor alpha (TNF-α) genes in disease-derived versus control cells
K. Okamura et al. 2017 [16]	38 patients with full thickness rotator cuff tear, sub-grouped according to tear size (no patient with small tears)	Glenohumeral Joint fluid	IL-1b; IL-6; IL-8	IL8 significantly correlated with resting pain IL-1b marginally not significantly correlated with resting pain
C. Shih et al. 2017 [17]	42 patients with chronic rotator cuff tear (21with partial thickness tears, 10 with non-massive, and 11 with massive tears)	Shoulder joint synovial fluid	IL-1b; IL6; MMP-1; MMP-13	MMP-1 and MMP-13 levels highest in the massive full-thickness group. IL-1β levels positively correlated with VAS (and functional scores)
C. Cho et al. 2021 [18]	63 patients 3-arm (21 with cuff tear and sleep-related pain; 21 with tear but normal sleep; 21 control group)	Peripheral blood sample	IL-1a; IL-1b; IL-2; IL-6; IL-8; IL-10; TNF-a But IL-1a, IL-1b, and IL-2 excluded from study	TNF-a higher in sleep disturbance group IL-8 and IL-10 higher in sleep disturbance group but no significant difference with normal sleep group

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
