# Peer review of "Biomolecules Related to Rotator Cuff Pain: A Scoping Review"

_biomolecules, 2022, doi:10.3390/biom12081016_

Round 1
Reviewer 1 Report
Thank you for giving me the opportunity to review for your journal. Below is a point by point review of the manuscript.
Topic: good; rotator cuff pain is the most common gray area of shoulder subspeciality.
Title: good
English: good
Abstract: good
Key words: good
Introduction: good
Method: please decide on a type of review and adhere to it; you cannot say you performed a scoping (narrative review) and describe an incomplete metanalysis. A good source of manuscript structuring can be found at http://www.prisma-statement.org/Extensions/ScopingReviews.
Results: good.
Discussion: good.
Conclusion: good
References: good
Tables and pictures: good
Overall: A good manuscript with pertinent topic and appropriate structure. Discretionary changes can apply.
Author Response
We would like to thank the reviewer for the comments, we appreciate the effort and time taken to review this article. Below a point by point response is given:
Method. We have structured this analysis as a scoping review, for reasons discussed in the manuscript (mainly large heterogeneity of assessed biomarkers and variation in types of samples taken from patients). This is not a narrative review nor a systematic one, the authors strictly followed the PRISMA guide for scoping reviews. The checklist, as stated in the text could be found in Appendinx A, we will upload it also below. Meta-analysis in not applicable to this article as there is a large heterogeneity in the assessed biomarkers (hence the scoping review). Once again thank you for taking the time to review this article.
Reviewer 2 Report
I think the content is sufficient.
The rotator cuff pain is of great clinical significance and is very well summarized in this review.
I think this is a very significant paper.
I have no special remarks.
Author Response
We would like to thank the reviewer for the comments, we appreciate the effort and time taken to review this article.
Reviewer 3 Report
Rotator cuff tear is a clinically important issue. I totally agree it is worth a scope review. I would like to provide several points.
First, the introduction should be strengthened. For example, the connection of sarcopenia with rotator cuff diseases should be mentioned (https://pubmed.ncbi.nlm.nih.gov/34026779/). Furthermore, in patients with diabetes and rotator cuff tears, glycation end products may be associated with shoulder range of motion limitation (https://pubmed.ncbi.nlm.nih.gov/35317765/).
Second, some subtitles are needed in the method portion. By the way, can the authors show any checklist regarding how to comply with certain guidelines for a scope review?
Third, although the PRISMA flow diagram has been used, no citation is provided in the method portion.
Fourth, the full terms for the abbreviations in the tables should be provided.
Fifth, a graphic abstract is suggested to add for clarification of the association of relevant biomolecules with rotator cuff tears.
Author Response
We would like to thank the reviewer for the comments and suggestions. We hereby provide a point by point answer to all comments.
- The introduction relates directly to rotator cuff related pain. The authors believe that expanding its scope to include other rotator cuff disease characteristics will weaken the introduction and make it less succinct and clear.
- Subtitles to the methods section have been added. As stated in the manuscript, a checklist was provided as Appendinx A.
- A reference for the flow diagram has been added.
- The full terms of all abreviations in this article are already provided in table 1.
- A graphic abstract has been created
This manuscript is a resubmission of an earlier submission. The following is a list of the peer review reports and author responses from that submission.
Round 1
Reviewer 1 Report
The problem of pain about the shoulder is complex and has a huge societal impact. The authors focus on the rotator cuff, which may be related to pain experienced by patients. The authors present an " update". How should this be interpreted, as a systematic review on evidence, as a narrative review or as an opinionated update.
For that matter some methodological issues which should be addressed: Please give details on literature Search (e.g. which databases -PubMed- language, Mesh headings etc). give a flow charts how many abstracts were found, and how many were deleted for which reason; did 2 reviewers perform the abstract selection? Please use a quality assessment tool for judging the quality of the abstracts.
If the above information is not available, the interesting information found by the authors, can only be viewed as a narrative or an opinion on an update, thus having selection bias a major concern. Data presented in the table are very heterogenous (both for location of sampling: bursa or glenohumeral joint; as well as for which cytokine). Thus conclusions drawn are difficult to interpret, due to bias.
Reviewer 2 Report
This review aims to analyse human studies that investigated for biomolecules found in synovial fluid, bursal or tendon tissues in patients suffering from rotator cuff related pain. While this is an interesting topic, the authors should propose a systematic review on that topic.
There is a high risk of missing information. Narrative review is useful when experts provide summary of evidence on a broad topic. Here, it should be a critically appraised topic at the best.
Reviewer 3 Report
please see editorial comments:
The manuscript presents an interesting point of view into cuff related shoulder pain. I think the manuscript in worthy for publication in a substantially revised form.
The authors must decide and make it clear in the title and abstract what type of review style article they chose: narrative/ scoping review, systematic review or meta-analysis. It is my assumption it is a narrative/ scoping review. Especially the beginning of the abstract is misleading and may create the impression of an original article.
Even for this simple review format there are several guidelines available to help structure the manuscript and help readers; the most common are the PRISMA and I recommend the authors use these (http://www.prisma-statement.org/Extensions/ScopingReviews).
The manuscript may be structured: 1. Intro, 2. Method: describe what you did and how you did it so that it can be reproduced. 3+ Results and discussions can be joined as it is now and presented as tendinopathy and tears (4.); finaly 5. Conclusions
Finally, a review is built on already available and published sources = references. Hence a review value can be judged by the quality and number of references. I recommend the authors to restructure their manuscript so that it gains roughly 20 more citations.
Thank you. Sincerely,
Reviewer 4 Report
This paper reviews the biomolecule associated with rotator cuff pathology.
Most of the content mentioned in this paper is already well known, and it is simply a description of the results of previous papers. Since this review article does not comprehensively describe the pathophysiology of rotator cuff disease, it is not of great help to clinicians.